

# A space and time efficient convolutional neural network for age group estimation from facial images

Ahmad Alsaleh and Cahit Perkgoz

Department of Computer Engineering, Eskisehir Technical University, Eskisehir, Turkey

## ABSTRACT

**Background**. Age estimation has a wide range of applications, including security and surveillance, human-computer interaction, and biometrics. Facial aging is a stochastic process affected by various factors, such as lifestyle, habits, genetics, and the environment. Extracting age-related facial features to predict ages or age groups is a challenging problem that has attracted the attention of researchers in recent years. Various methods have been developed to solve the problem, including classification, regression-based methods, and soft computing approaches. Among these, the most successful results have been obtained by using neural network based artificial intelligence (AI) techniques such as convolutional neural networks (CNN). In particular, deep learning approaches have achieved improved accuracies by automatically extracting features from images of the human face. However, more improvements are still needed to achieve faster and more accurate results.

**Methods**. To address the aforementioned issues, this article proposes a space and time-efficient CNN method to extract distinct facial features from face images and classify them according to age group. The performance loss associated with using a small number of parameters to extract high-level features is compensated for by including a sufficient number of convolution layers. Additionally, we design and test suitable CNN structures that can handle smaller image sizes to assess the impact of size reduction on performance.

**Results**. To validate the proposed CNN method, we conducted experiments on the UTKFace and Facial-age datasets. The results demonstrated that the model outperformed recent studies in terms of classification accuracy and achieved an overall weighted F1-score of 87.84% for age-group classification problem.

Corresponding author
Cahit Perkgoz, cahitperkgoz@eskisehir.edu.tr

# INTRODUCTION

Facial images possess various characteristics such as age, gender, expression, social status, and ethnicity (*Alley, 2013*). If these features are accurately estimated, they can be utilized in numerous domains ranging from social life to technology. It is crucial to employ these facial features with scientific methods and apply them in real applications and commercial areas. Among these features, age estimation is one of the most critical, and can be highly beneficial

in real-time applications such as age simulation, age-based access authorization, finding missing persons, electronic customer relationship management, and age-specific human–computer interaction (*Al-Shannaq & Elrefaei, 2019*). Additionally, age estimation is viewed as a complementary and flexible biometric as it provides supplementary information about the user's identity, along with biometric features such as fingerprint and iris. Furthermore, there may be age-related restrictions for logging into a website or mobile application. However, age estimation from a single picture of an individual has not yet been adequately estimated (*Agbo-Ajala & Viriri, 2021*).

Although several methods, including classification-based (*Geng, Zhou & Smith-Miles, 2007*), regression-based (*Shen et al., 2018*), ranking-based (*Yang, Zhong & Metaxas, 2010*), hybrid-based (*Guo et al., 2008*), and soft classification methods (*Alnajar et al., 2012*), have been developed for age estimation, they can be broadly categorized into hand-crafted models and artificial intelligence models. The primary distinction is that artificial intelligence algorithms automatically extract features with minimal human intervention, instead of manual extraction. They continuously learn from existing data through experience and have been applied to numerous real-world problems. These algorithms have provided innovative solutions, particularly in solving complex problems. With the rise in datasets in recent years, these low-cost algorithms, especially deep learning models, have been widely used (*Jordan & Mitchell, 2015*).

A considerable amount of research has been conducted in the field of automatic age estimation to extract features and estimate age using methods apart from handcrafted ones (*Agbo-Ajala & Viriri, 2021*). In 2015, a three-layer convolutional neural network was developed (*Levi & Hassner, 2015*) to represent facial features, using regularization techniques to reduce overfitting on a small set of images. Despite the simplicity of its architecture, the proposed CNN surpassed state-of-the-art approaches at the time. That same year, an end-to-end learning method called AgeNet was presented (*Liu et al., 2015*), which combined a classification model based on Gaussian label distribution with a regression model based on the actual value.

A deep CNN with a Gaussian loss function for age estimation was proposed (*Ranjan et al., 2015*) to perform the age estimation task from unconstrained facial photos. In the model, face detection and face alignment were performed on the pictures, and then a 3-layer CNN was applied to reveal and classify facial features. The model achieved better performance than the conventional linear model for age estimation.

A method using two deep CNNs with distribution-based loss functions running in parallel was published (*Huo et al., 2016*). One of the two architectures was transferred from VGG-16 and fine-tuned with three different datasets. Meanwhile, the second architecture was an innovative CNN structure trained with public datasets and a variety of additional datasets. The results were combined, and relatively more accurate results were obtained. In another study, (*Antipov et al., 2016*) a VGG-16 based CNN model was trained by the massive IMDB-WIKI dataset. The network was divided into children and general networks that were fine-tuned with different age encoding approaches. A strict age encoding was applied to the child network, and label distribution encoding to the general network.

The successful results obtained in this study won the first prize in the ChaLearn LAP age estimation competition in 2016.

An optimized CNN with four convolutional and two fully connected (FC) layers was suggested (*Aydogdu & Demirci, 2017*) using MORPH-II dataset. The performance of the results was evaluated using the "exact success", "top-3", and "1-off" criteria, and it outperformed the existing CNN designs.

In their study *Chen, Zhang & Dong (2017)*, presented a series of CNNs, called ranking-CNN, for age estimation. The final age estimate is determined by the binary output of each CNN. This architecture has significantly better performance than other existing multi-class architectures applied to the same dataset.

VGG-Face network and GoogLeNet are employed separately in a deep CNN with eight convolutional layers followed by three FC layers for age estimation (*Qawaqneh, Mallouh & Barkana, 2017*). Rectified linear units are used at each convolutional layer, and a max-pooling operation is applied to each output of a layer. Despite being trained with a huge amount of data, the GoogleNet was outperformed by the VGG-Face network.

A novel CNN method called residual networks of residual networks (RoR) has been introduced (*Zhang et al., 2017*) for age grouping and gender estimation. The network was pretrained on the ImageNet dataset and fine-tuned by the IMDB-WIKI-101 dataset. The performance of the network was evaluated on different benchmarks, and impressive results were achieved.

Another deep learning approach developed using the VGG-16 architecture is called Deep EXpectation (DEX) (*Rothe, Timofte & Van Gool, 2018*). In this study, actual and apparent age were estimated from a single face image without using facial marks. The network was pretrained with labeled IMDB-WIKI and ImageNet datasets. Prior to the expected value for age regression, a robust face alignment is applied to DEX, yielding state-of-the-art results for actual and apparent age.

Age estimation has also been used as an auxiliary method in solving a high-level problem (*Savchenko, 2019*). The author took advantage of age estimation using public convolutional networks to organize photographs, with advantageous results in terms of runtime and space complexity.

In *Nam et al. (2020)*, the authors introduced a deep learning-based solution for low-resolution face images. They first obtained high-resolution images from low-resolution ones using generative adversarial networks (GAN). Then, they provided the enhanced images to CNN network architectures such as ResNet, VGG, and DEX for training. The trained networks were tested with unseen data, and they generated effective results while estimating the ages.

A recent study (*Agbo-Ajala & Viriri, 2020*) was published to classify age groups and genders of real images. The images were first preprocessed and converted into suitable inputs for the CNN algorithm. The features were extracted using the proposed CNN algorithm, and classification was made according to age group and gender. Test experiments were applied to the OIU-Adience dataset, and better results were obtained in this dataset compared to previous studies.

A hybrid approach (*Rwigema, Mfitumukiza & Tae-Yong, 2021*) using artificial neural networks and CNN by using decision fusing techniques is proposed and applied to two different datasets where images are divided into four different age groups. By using these two datasets, namely Adience benchmark database and purposely produced data for age estimation, accuracy results of 84.6% and 86.1% are obtained, respectively.

A transfer learning method using different pre-trained networks is presented for several number of groups determined according to the selected age range (*Dagher & Barbara, 2021*). The best results were obtained using GoogleNet with accuracy rates of 74% and 89% when the dataset was divided into groups of 6 and 3, respectively.

A recent study (*Rizwan et al., 2021*) published a multi-step method for age group classification by preprocessing images and using a CNN. After extracting facial landmark features, a CNN is used for classification. The method was applied to three different datasets and produced performance values over 90%. However, the proposed method consists of many steps and requires a high number of calculations. Moreover, the used datasets were not consistently grouped and could include multiple facial images of the same person.

Another method (*ELKarazle, 2022*) proposes the use of generative adversarial networks to improve the quality of images before feeding them into a CNN for age group classification. It has been shown in this study that the use of GANs in combination with CNNs enhances performance for a number of age grouping problems. The proposed method was applied to different problems, and an accuracy of 80.16% was achieved in the classification problem with four classes.

A CNN with four layers with more than one million parameters was proposed for age and gender estimation (*Sharma, Sharma & Jindal, 2022*). When applied to the UTKface dataset, the performance results were given as a mean absolute error of 0.77 for the age regression problem and an accuracy of 99.86% for gender classification.

More recently, a two-stage method (*Raman, ELKarazle & Then, 2022*) for sequential gender classification and age group estimation was proposed. In their study, images with male and female faces were distinguished using two models based on VGG16 networks, trained on separate male and female sets. Age estimation models were then trained and tested on the UTKFace and FG-NET datasets for four age groups, and it was observed that using two different models improved overall accuracy.

While significant progress has been made in age estimation with deep learning algorithms, this problem still needs improvement. CNNs are the most promising approach for image processing, and previous studies have shown that new CNN structures, training methods, and data preprocessing can improve performance. However, designing deep networks with a large number of parameters can cause overfitting and lead to performance degradation on unseen data. Additionally, performance is directly proportional to the amount of data in the training set. Therefore, to achieve better results, it is crucial to develop well-designed CNN structures and work with larger datasets.

In this study, two commonly used datasets were combined and augmented to increase the amount of training data, while the number of parameters was kept low to reduce simulation time and avoid overfitting. The proposed CNN structure achieves a significant reduction in the number of parameters by utilizing few and small-size filters. It employs

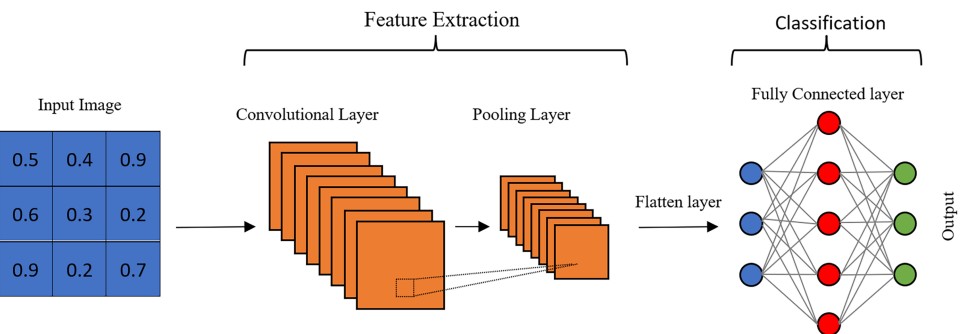

**Figure 1** **Architecture of a CNN.**

sufficient number of layers to compensate for the reduced number of parameters and extract high-level features. This new deep CNN structure outperformed previous studies by producing more accurate results in less time, offering a promising solution for age estimation with fewer parameters.

## MATERIALS & METHODS

Research on age estimation to date has shown that convolutional neural networks can increase the accuracy of estimations compared to other approaches. They can directly learn the features required for age estimation from facial images. In fact, CNNs were specifically developed to draw meaningful results from image data (*Liu et al., 2015*; *Antipov et al., 2016*). A CNN is an end-to-end layered structure that is independent of prior knowledge, which is the most significant benefit compared to existing hand-crafted methods. Apart from the input and output layers, it essentially has a convolution layer, a pooling layer, and a fully connected layer (Fig. 1). The number of feature extraction layers can be increased in order to extract high-level features in deep CNN structures.

The convolutional layer is a crucial component that sets CNNs apart from traditional neural networks. It applies multiple filters to images by sliding over them to extract features. As the filter moves, it creates an activation map on the image. While the feature extraction process is similar to other neural networks, the number of parameters is significantly reduced in CNNs by using the same filter with limited spatial dimensions in different regions. This is a significant advantage for processing large-scale data such as images. By reducing the number of parameters, the network also mitigates the risk of overfitting (*Qiu, 2016*). Each filter performs a convolution on the input image, and a bias term is added, as shown in Eq. (1), to reveal a specific feature.

$$h^{k+1} = \sigma(W^k \otimes h^k + b^k) \qquad (1)$$

where $\otimes$ is refers to the convolution operation, $h^k$ is the $k$th input feature map of the convolution layer, $W^k$ is the weight matrix of the filters, $b^k$ is the bias, $h^{k+1}$ is the output feature map, and $\sigma(.)$ is an activation function. The output of the convolution layer is a stack of feature maps with a depth that depends on the number of filters applied. Each filter

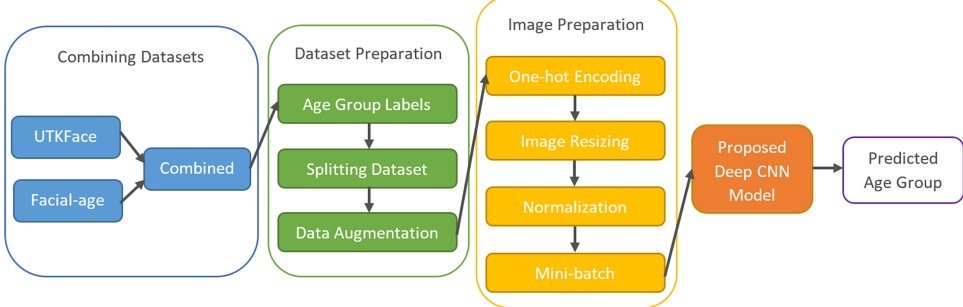

**Figure 2** The framework of the proposed age group classification model.

reveals low-level features such as edges, colors, and orientations of objects on the input image in the first layers. As the network deepens with additional layers, high-level features are extensively revealed.

The main purpose of the pooling layer is to reduce the spatial size of the convolution output without losing features that are not dependent on position and rotation (*Li, Xie & Li, 2017*). Size reduction directly reduces the number of computations in subsequent layers, thus speeding up the algorithm. In addition, pooling provides positional stability to the network. The most common pooling operations are max-pooling and average pooling. Max-pooling selects the highest value in a patch. Therefore, while it is advantageous in terms of capturing the extreme values of certain features, it is also possible to capture undesirable noise values. Average pooling evaluates the average of the values in a certain patch and may reduce the discriminative power, while it may be useful in reducing noise.

The combination of a convolution layer and pooling layer can be counted as the $i$th feature extraction layer of a CNN. The number of layers can be increased to capture high-level features, but this requires more computational power. The final output of these layers is flattened and given as input to the FC layer. Unlike the convolutional layer, the neurons of the next layer are fully connected to all the neurons of the previous layer. In other words, the FC layer is a fully connected layer without weight sharing, and each filter of this layer has a size of $1 \times 1$.

Another important aspect of a neural network is the activation functions. Sigmoid, hyperbolic tangent, and rectified linear unit are frequently used activation functions for each artificial neuron, and their most important purpose is to increase the nonlinearity of the networks. At the same time, an activation function increases the network stability by generating feature maps free of excessive data values.

Figure 2 shows the framework of the proposed age-group classification model. The model, which basically consists of preparing the appropriate inputs for the CNN and classifying the samples by extracting the facial features through the CNN architecture, is explained step by step in the following.

The unconstrained dataset contains more than 33,000 images with $200 \times 200$ pixels created by combining the UTKFace (*Zhang, Song & Qi, 2017*) and Facial-Age (WIKI_ART https://www.kaggle.com/datasets/frabbisw/facial-age) datasets. The images have various

**Table 1  Each age group with the corresponding class (label) and number of images in each dataset.**

| Age group | Class (label) | UTKFace | Facial-age | Combined | Percentage (%) |
|---|---|---|---|---|---|
| 0–2 | 0 | 1,605 | 1,587 | 3,192 | 9.53 |
| 3–7 | 1 | 1,028 | 964 | 1,992 | 5.95 |
| 8–14 | 2 | 1,018 | 916 | 1,934 | 5.78 |
| 15–20 | 3 | 1,226 | 800 | 2,026 | 6.05 |
| 21–27 | 4 | 5,572 | 1,119 | 6,691 | 19.98 |
| 28–45 | 5 | 7,646 | 1,710 | 9,356 | 27.94 |
| 46–65 | 6 | 3,914 | 1,684 | 5,598 | 16.72 |
| 66+ | 7 | 1,699 | 998 | 2,697 | 8.05 |

**Table 2  The number and the percentage of images in separated datasets.**

| Dataset | Size (%) | Number of images |
|---|---|---|
| Training dataset | 70 | 23,440 |
| Validation dataset | 22.5 | 7,534 |
| Test dataset | 7.5 | 2,512 |
| Combined dataset | 100 | 33,486 |

occlusions and expressions and were obtained in natural environments under different lighting conditions. Each image in these datasets is a colored picture in JPG and PNG format, and age labels are added along with some other labels. Since age analysis was the focus of this study, only age labels of the images were used. The age range is between 0 and 116, and the images have different facial expressions, poses, illuminations, and resolutions.

The dataset preparation phase began with the labeling of age groups. The dataset was divided into eight basic groups: 0-2, 3-7, 8-14, 15-20, 21-27, 28-45, 46-65, and 66+. Table 1 shows the number of images and the percentages of each age group in UTKFace, Facial-age, and the combined dataset.

As a standard procedure in the development of artificial intelligence methods, datasets are typically divided into three subsets: training, validation, and test datasets. During this division, each subset is partitioned to ensure that the class ratios are maintained. Table 2 displays the number and percentage of images in each subset.

Increasing the amount of data used for training neural networks can improve the quality of the results, despite the longer training time required. To achieve this, the training dataset was augmented by creating additional images that increase variance and prevent overfitting. Nine new images were generated for each existing image by reversing and rotating them at different angles. This method increased the size of the training dataset by a factor of 10.

Each sample in the dataset is accompanied by a label. To perform computer operations, categorical labels must be converted into numerical values ranging from 1 to 8. However, one-hot decoding can be a more effective method, which involves a binary vector to represent the eight labels. For instance, a sample in the 5th group can be represented by a vector with a "1" at the 5th position, like [0 0 0 0 1 0 0 0].

**Table 3  Number of parameters used in most common CNN architectures (*Keras Models, n.d.*).**

| Model | # Of Parameters |
| --- | --- |
| Xception | 22.9M |
| VGG16 | 138.4M |
| VGG19 | 143.7M |
| ResNet50V2 | 25.6M |
| ResNet101V2 | 44.7M |
| ResNet152V2 | 60.4M |
| InceptionV3 | 23.9M |
| MoblieNetV2 | 3.5M |

The deep CNN architecture was originally designed for images in the combined dataset with dimensions of 200 × 200 pixels. To evaluate the effects of reducing image dimensions, additional experiments were conducted using smaller sizes. These experiments will be referred to as Case-I, Case-II, Case-III, and Case-IV for images with dimensions of 200 × 200, 100 × 100, 50 × 50, and 25 × 25 pixels, respectively.

Normalization of image pixel values is necessary to stabilize optimization techniques and achieve faster convergence. To accomplish this, the images were subjected to min-max normalization as described in Eq. (2), which converts the pixel values into the range of [0, 1].

$$u_{\text{scaled}} = \frac{u_{\text{input}} - u_{\text{min}}}{u_{\text{max}} - u_{\text{min}}}. \tag{2}$$

Following the normalization process, the data was divided into small batches rather than feeding all of it to the CNN at once. Each batch contains a certain number of images, which is determined by a hyperparameter. Using mini-batches speeds up the algorithm and enables it to work with less computational power.

CNN models described in the literature contain a large number of adjustable parameters, as shown in Table 3, which presents the most successful studies and the number of parameters used. The performance of a neural network increases up to a point as the number of parameters increases, but this also increases the duration of the training process. In the absence of high processing power, the training process can take days or even weeks. Additionally, using a large number of parameters can lead to overfitting problems. Although CNNs require fewer weights than fully connected neural networks, this study reduced the size and number of filters to further decrease the number of parameters. However, the number of layers was increased to uncover high-level features.

The proposed CNN model operates in two phases: feature extraction and age group classification. The first stage comprises seven convolution layers, each with 32, 32, 64, 64, 128, 128, and 256 filters, respectively. The filters were initialized using the Glorot Uniform initialization method outlined in Eq. (3) (*Glorot & Bengio, 2010*). This technique depends on the number of connections between two layers and enables the algorithm to converge

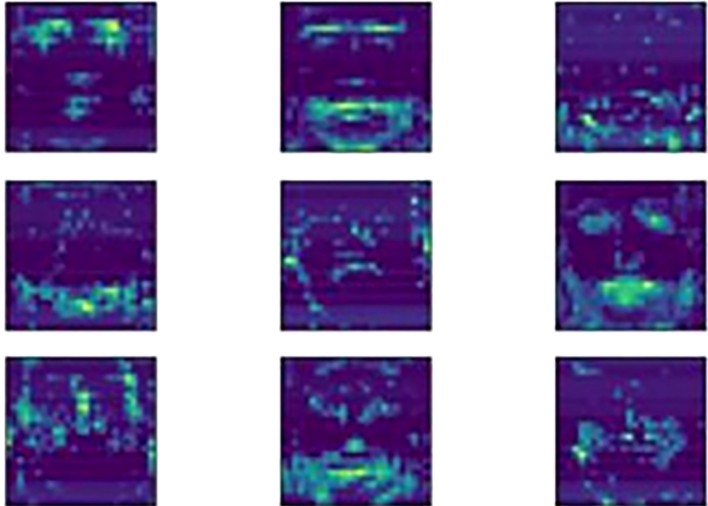

**Figure 3** Output feature map visualization.

more quickly while preserving activation variances during the training process.

$$W \sim U\left[ -\frac{\sqrt{6}}{\sqrt{n_j + n_{j+1}}}, \frac{\sqrt{6}}{\sqrt{n_j + n_{j+1}}} \right]. \tag{3}$$

Appropriate padding numbers are used to produce the outputs with the same size of the inputs in the corresponding convolution layer. Pooling operations were applied for size reduction after two convolution layers to carry the image up to the last layer so that the size of the images would not vanish. In this study, the flattening layer is provided by the global max pooling operation where the most important information of a feature is transferred to the classification layer and redundant information is blocked. Figure 3 shows some examples of the features of an input image extracted from the last pooling layer. These features appear to perceive details and are sufficiently distinguishable.

The next stage is the prediction (classification) of age groups using the extracted features. Two fully connected layers are designed with 128 and eight neurons, respectively. Rectified linear unit (ReLU) activation functions are used in all layers of the CNN model except the output layer, where the activation function is a softmax transfer function. This function assigns a probability to each age group prediction, with a categorical cross-entropy loss function (Eq. (4)) used to evaluate the performance of the model:

$$-\sum_{i=1}^{N} T_i \log(p_{o,i}) \tag{4}$$

where $N$ is the number of age-groups and $T_i$ is the binary indicator where it is 1 for the largest $p_{o,i}$, and 0 for the rest. If observation $o$'s actual class label is $i$, $p$ is the predicted probability observation $o$ of class (age-group) $i$.

Each layer of the proposed CNN architecture and its output image size, along with the number of parameters for that layer, are given in Table 4 for Case-I. A total of 618,940

**Table 4  Proposed deep CNN architecture.**

| Layer (type) | Output size | Number of parameters |
|---|---|---|
| Input Image | 200 × 200 × 1 | – |
| Conv 1 | 200 × 200 × 32 | 320 |
| BatchNormalization 1 | 200 × 200 × 32 | 800 |
| Conv 2 | 200 × 200 × 32 | 9,248 |
| BatchNormalization 2 | 200 × 200 × 32 | 800 |
| Max-pool 1 | 100 × 100 × 32 | – |
| Conv 3 | 100 × 100 × 64 | 18,496 |
| BatchNormalization 3 | 100 × 100 × 64 | 400 |
| Conv 4 | 100 × 100 × 64 | 36,928 |
| BatchNormalization 4 | 100 × 100 × 64 | 400 |
| Max-pool 2 | 50 × 50 × 64 | – |
| Conv 5 | 50 × 50 × 128 | 73,856 |
| BatchNormalization 5 | 50 × 50 × 128 | 200 |
| Conv 6 | 50 × 50 × 128 | 147,584 |
| BatchNormalization 6 | 50 × 50 × 128 | 200 |
| Max-pool 3 | 25 × 25 × 128 | – |
| Conv 7 | 25 × 25 × 256 | 295,168 |
| BatchNormalization 7 | 25 × 25 × 256 | 100 |
| Max-pool 4 | 12 × 12 × 256 | – |
| Global Max-pool 1 | 256 | – |
| FC 1 | 128 | 32,896 |
| BatchNormalization 8 | 128 | 512 |
| FC 2 | 8 | 1,032 |
| Total params: 618,940, Trainable params: 617,234, Non-trainable params: 1,706 | | |

trainable parameters are used for 200 × 200 images, which is much less compared to the other studies given in Table 3. The first layer of the CNN is set to the same dimensions as the provided input images. In this study, we also performed experiments for the other three cases and adjusted the parameters of each input layer. If the structure in Table 4 is used for Case-III and Case-IV, there will be no pixels containing meaningful information when an image reaches the last pooling layer. Therefore, the last one and the last two layers were removed from the initial model for Case-III and Case-IV images, respectively.

## RESULTS

The proposed deep CNN models were executed on a computer with an Intel Core™ i7-6700HQ CPU Processor, 12GB RAM and NVIDIA RTX 2060 6GB GPU with the combined dataset. The data were divided into eight different classes according to the age ranges of 0–2, 3-7, 8–14, 15–20, 21-27, 28-45, 46–65, and 66+. Images provided from the UTKFace and Facial-age datasets were in color and 200 × 200 pixels in size. They were converted to grayscale images and split into training, validation, and test data at the percentages previously specified in Table 2. To observe the effects of reducing the

dimensions of the images in terms of performance and time, the images were resized to 100 × 100, 50 × 50, and 25 × 25 pixels. The CNN model was run with the appropriate number of layers for Case-I, Case-II, Case-III, and Case-IV. The parameters (weights) were initialized with Glorot_Uniform at the beginning, and the Adam optimizer was utilized to optimize the loss function with a learning rate of 0.001 and mini-batches of size 50 during training for 100 epochs.

The classification performance of this model was evaluated through a number of empirical experiments with different evaluation metrics, which include the confusion matrix, accuracy, precision, recall, and F1-score.

The **confusion matrix** is a table presenting the predicted and actual classes. Since there are eight age groups in the problem, the table consists of eight columns and eight rows where each column and row consist of the predicted and actual number of samples, respectively.

**Accuracy** is the most commonly used performance measure when the dataset is balanced. It returns the ratio of correctly classified face images to the total number of dataset images (Eq. (5)).

$$\text{Accuracy} = \frac{TP + TN}{TP + TN + FP + FN} \times 100 \tag{5}$$

where TP is the number of true positive predictions, TN is the number of true negative predictions, FP is the number of false positive predictions, and FN is the number of false negative predictions.

**Precision** is the percentage of correctly classified predicted images (Eq. (6)). It is used for datasets that are unbalanced according to the number of samples in the classes. A higher precision score indicates that the model is recognizing the age group properly.

$$\text{Precision} = \frac{TP}{TP + FP}. \tag{6}$$

**Recall** is related to sensitivity and gives an idea of the ratio of correctly predicted age groups and all predicted positives (Eq. (7)). It puts more emphasis on increasing the number of accurately predicted examples in a certain class.

$$\text{Recall} = \frac{TP}{TP + FN}. \tag{7}$$

**F1-score** combines precision and recall, measuring the overall accuracy and performance of a model. A high F1-score implies that a model effectively predicted age groups while minimizing false positives and false negatives (Eq. (8)). As mentioned before, the combined dataset is not balanced, which would bias the model towards certain age groups rather than others. Therefore, the F1-score would be more decisive in analyzing the performance of the CNN on the combined dataset in this study.

$$\text{F1\_score} = \frac{\text{precision} \times \text{recall}}{\text{precision} + \text{recall}} \times 2. \tag{8}$$

While training a model, generally, accuracy metrics and loss function values are observed. As the accuracy values increase up to a point during the learning process of the model,

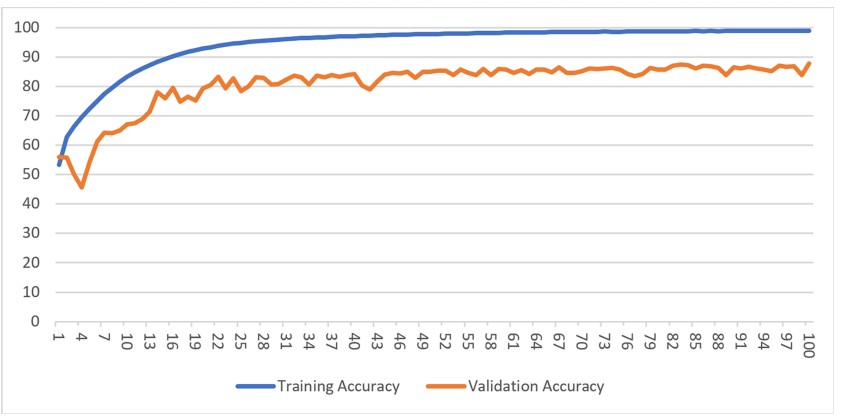

**Figure 4** Training and validation accuracy for Case-I.

**Table 5 Average and total training time for each experiment.**

| Experiment | Number of epochs | Average time (s) | Total Training time (s) |
|---|---|---|---|
| Case-I | 100 | 1,000 | 100,000 |
| Case-II | 100 | 310 | 31,000 |
| Case-III | 100 | 204 | 20,400 |
| Case-IV | 100 | 186 | 18,600 |

the loss values decrease and then stabilize. When there are no further improvements, or the accuracy of the validation dataset begins to deteriorate, the training process is stopped. Figure 4 shows the accuracy and loss of the model with an image resolution of $200 \times 200$ pixels for both the training and validation datasets in Case-I.

Likewise, the training process was completed for the datasets with reduced image sizes in Case-II, Case-III, and Case-IV, and the training process was stopped when similar trends were observed. Table 5 shows the average and total training time for 100 epochs for datasets with different sizes of images.

After the training of the network is completed, the performance of the model can be examined with the test data that has not been used during the training phase. All classes in the test dataset are unbalanced. Therefore, to get a better idea of the performance of the CNNs, the results should be analyzed not only with the accuracy metric but also with the confusion matrix, precision, recall, and especially F1-score metrics.

The classification results of the CNN model for Case-I were obtained with an overall accuracy of 86.58% for all classes. Figure 5 shows the results of the confusion matrix measure of our model's performance on the test dataset. It is seen that the 0-2, 3-7, 8-14, and 66+ age groups are classified with an accuracy close to 90%. This is reasonable because the faces of people in these age groups have distinctive features that allow the model to learn more about them and distinguish these classes better than the others. At the same time, it can be observed that when the model misclassifies a certain class in the confusion

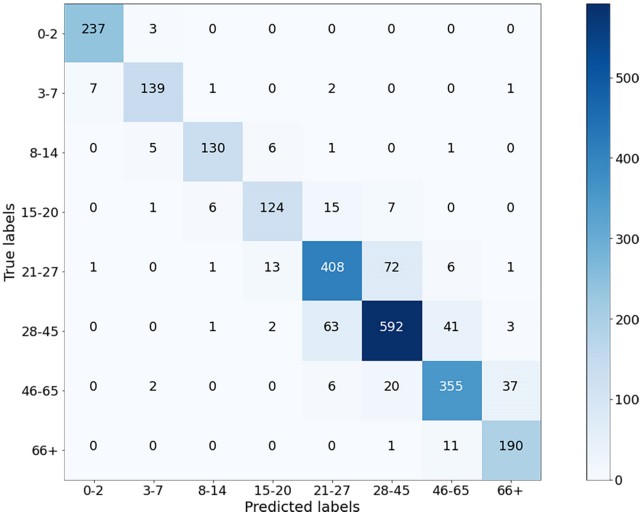

**Figure 5** Confusion matrix on test data for Case-I.

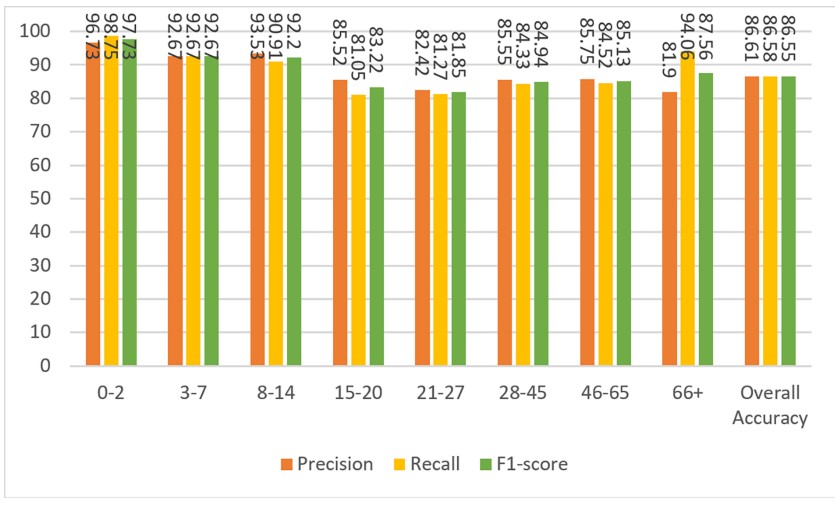

**Figure 6** Precision, recall, and F1-score metrics on test data for Case-I.

matrix, it misclassifies in favor of the neighboring class, and misclassifications of the model contain minor errors.

Figure 6 shows the results of the model's precision, recall, and F1-score measurements for the test data. Although the dataset is not balanced, it can be clearly seen that the results for almost every metric are close to each other and high. The overall metric results for the entire dataset can be seen in the last columns of each group in Fig. 6. A result of 86.55% was obtained for F1-score, which is the harmonic mean of precision and recall values and contains information from both.

Next, the images from the combined dataset were resized to lower dimensions (Case-II, Case-III, and Case-IV), and the training and testing procedures were repeated. The aim

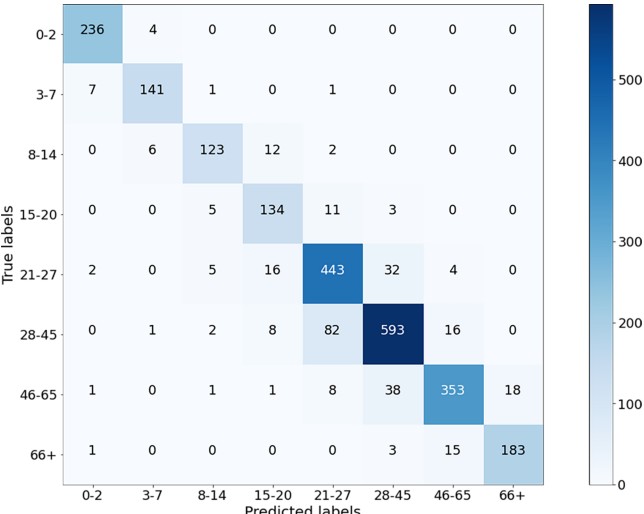

**Figure 7** Confusion matrix on test data for Case-II.

here is to observe the robustness of the proposed method and the variation in the accuracy values by decreasing the sizes. It is clear that for small sizes of images, the simulation will be faster as the algorithm will contain fewer parameters. If the performance of the network does not degrade too much, it is preferable to use the images in a reduced size. Figure 7 shows the results of the confusion matrix for Case-II. Based on these values, the precision, recall, and F1-score measurements of the model on the test data are given in Fig. 8. As can be seen, the obtained results are slightly better than the results of Case-I. A weighted F1-score of 87.84% was obtained for the overall dataset.

Later, the experiment was performed for Case III where the confusion matrix and performance measurements are given in Figs. 9 and 10, respectively. As can be seen, there are small decreases in performance values compared to the previous experiments. The overall performance value for weighted F1-score was obtained as 86.41%.

Finally, the experiment was performed for Case IV where the dimensions of the images were 25 × 25. It can be noticed that the performance has degraded compared to Cases I, II, and III, but is still acceptable. This is because facial features disappear as the image shrinks. The confusion matrices and performance measures of the experimental results can be seen in Figs. 11 and 12.

## DISCUSSION

In this section, the results are compared both among themselves and with previous studies. The performance measures calculated in the previous section for weighted overall data are tabulated in Table 6. Among them, the best results were obtained with 100 × 100 images (Case II) where the F1-score is 87.84%. Similar to human intelligence, artificial intelligence algorithms will have poor performance in extracting information from poor quality images. While the UTKFace and Facial age datasets are prepared for public use, the smaller images were enlarged to 200 × 200 pixels and blurred. Therefore, Case II produced better results

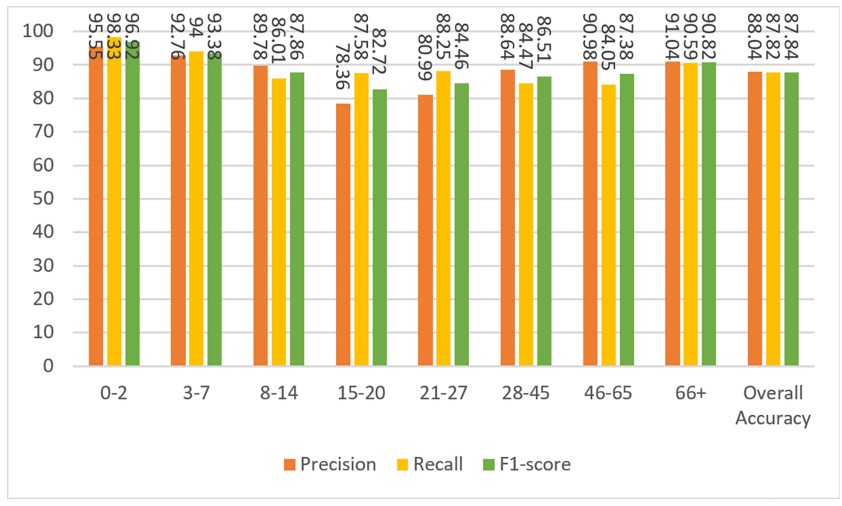

**Figure 8** Precision, recall, and F1-score metrics on test data for Case-II.

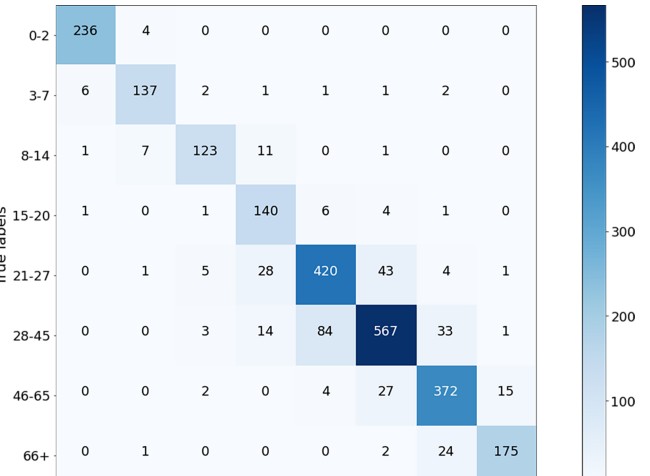

**Figure 9** Confusion matrix on test data for Case-III.

than Case I, as $100 \times 100$ images are closer to the original dimensions than $200 \times 200$ pixel images in Case I. On the other hand, Case III and Case IV experiments show comparatively worse results as some features are lost when the images are scaled down.

The best result of this study is also compared with some other works mentioned in the introduction that used the same methodology for age estimation task in Table 7. It should be noted that the proposed model outperforms all other previous works.

The performance of the presented algorithms has improved over the years. The best results were obtained in the study (*Agbo-Ajala & Viriri, 2020*) with an accuracy of 84.8% and an F1 score of 81.58%. The age ranges determined for age groups in their study are

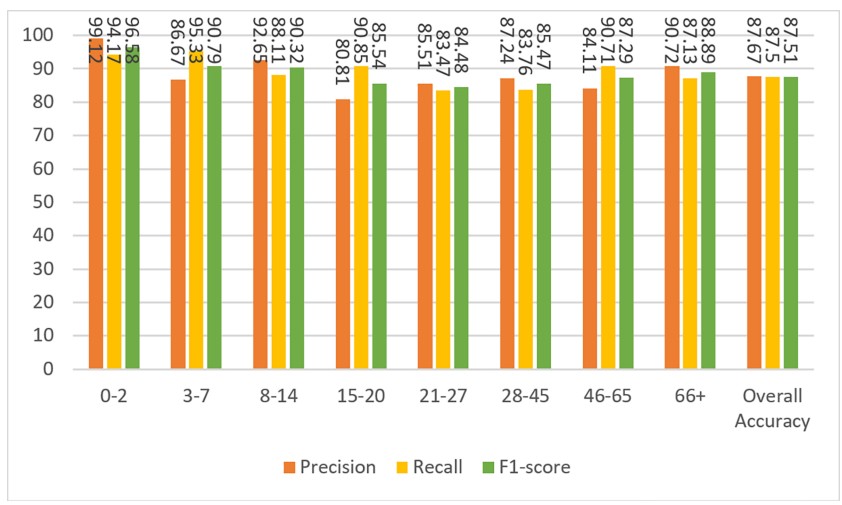

**Figure 10** Precision, recall, and F1-score metrics on test data for Case-III.

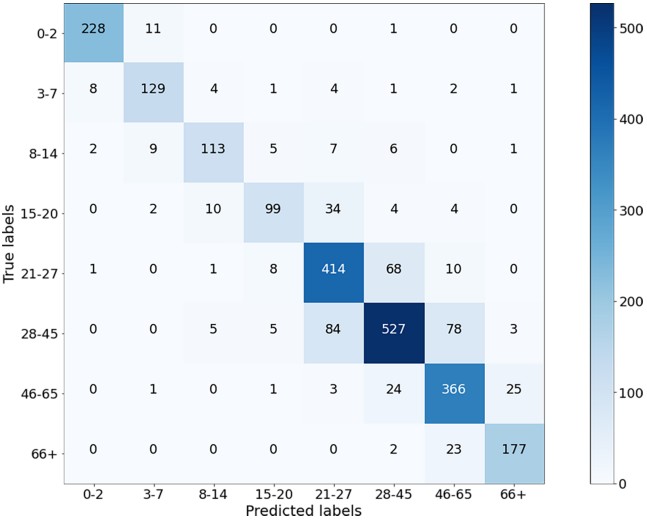

**Figure 11** Confusion matrix on test data for Case-IV.

close to the ranges defined in this study. Figure 13 shows the confusion matrix of their results for age groups of the OIU-Adience dataset, where the total number of images in the test data is 992.

Based on the confusion matrix given in Fig. 13, precision, recall, and F1-score measures are calculated for the test data and given in Fig. 14.

To compare the Agbo-Ajala and Viriri model and the model described here, the F1-score for all age groups and the overall data is tabulated in Table 8.

The overall F1-score of the Agbo-Ajala and Viriri model is 81.58%, which is much lower than the overall performance (87.84%) of the proposed model. In addition, Table 8

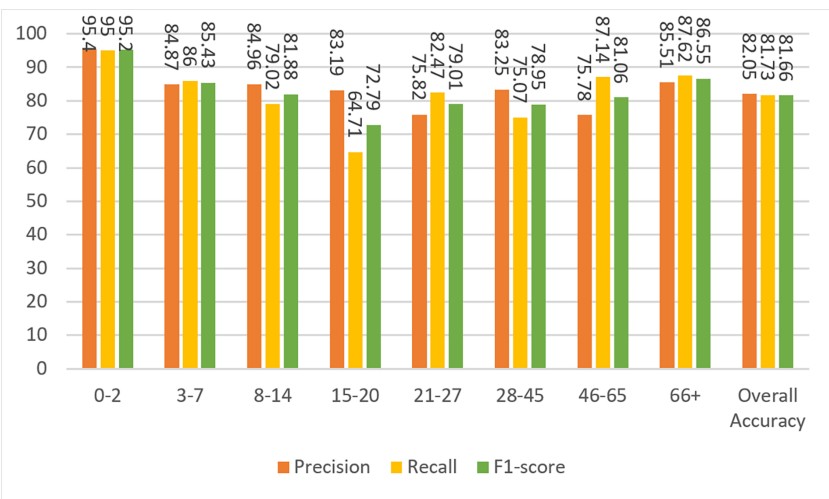

Figure 12 **Precision, recall, and F1-score metrics on test data for Case-IV.**

Table 6 **Performance metrics of the experiments.**

| Experiment | Accuracy (%) | Precision (%) | Recall (%) | F1-Score (%) |
|---|---|---|---|---|
| Case-I | 86.58 | 86.61 | 86.58 | 86.58 |
| Case-II | 87.82 | 88.04 | 87.82 | 87.84 |
| Case-III | 86.38 | 86.70 | 86.39 | 86.41 |
| Case-IV | 81.73 | 82.05 | 81.73 | 81.66 |

Table 7 **Comparison of the proposed work and previous studies with the same methodology.**

| Reference | Approach | Accuracy | F1-score | # Of Parameters |
|---|---|---|---|---|
| *Levi & Hassner (2015)* | CNN + dropout | 50.7% | – | – |
| *Aydogdu & Demirci (2017)* | 4C2FC | 46.4% | – | – |
| *Qawaqneh, Mallouh & Barkana (2017)* | VGG + dropout | 59.9% | – | – |
| *Zhang et al. (2017)*; *Zhang, Song & Qi (2017)* | RoR | 67.3% | – | – |
| *Rothe, Timofte & Van Gool (2018)* | DEX | 64.0% | – | – |
| *Agbo-Ajala & Viriri (2020)* | 4C2FC | 84.8% | 81.58% | 2.5M |
| Proposed Work | 7C2FC | 87.82% | 87.84% | 615K |

demonstrates that the F1-scores obtained from our model are superior to the Agbo-Ajala and Viriri model, except for the 21-27 and 66+ age groups. Also, as previously mentioned in Table 7, the proposed CNN model is about four times better when compared in terms of the number of parameters used. This saves a lot of computational resources and results in much faster training time.

Although it is scientifically pleasing that AI techniques produce successful results in many scientific fields, including this problem, it is important to be cautious about ethical concerns. First, AI systems can learn from biased data, which can lead to the spread of societal prejudices and discrimination. Such biases can be magnified by AI systems. Second,

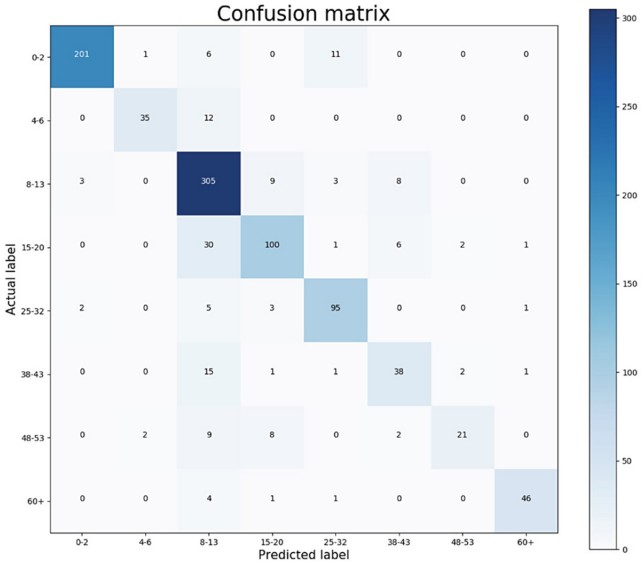

**Figure 13** Confusion matrix from the work of *Agbo-Ajala & Viriri (2020)*.

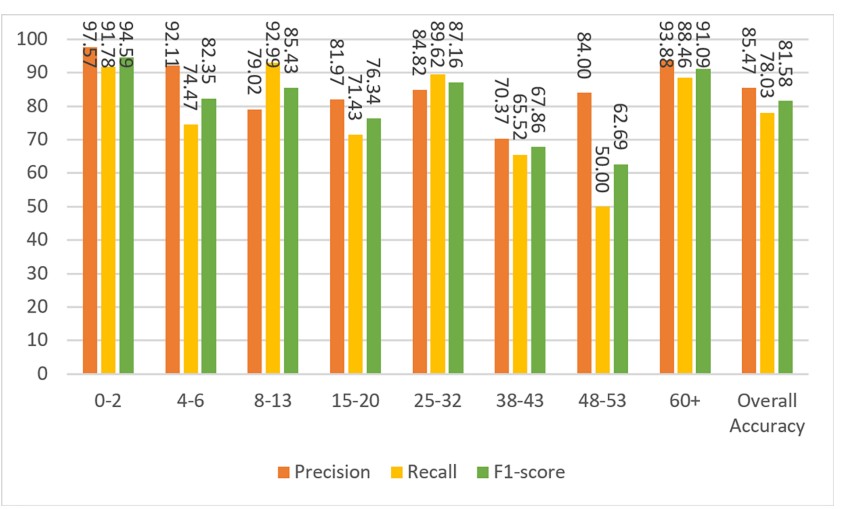

**Figure 14** Precision, recall, and F1-score metrics from the work of *Agbo-Ajala & Viriri (2020)*.

**Table 8** Comparison of proposed and (*Agbo-Ajala & Viriri, 2020*) model in terms of F1-score. (The numbers in bold indicate the age groups.)

|  | **0-2** | **3-7** | **8-14** | **15-20** | **21-27** | **28-45** | **46-65** | **66+** | **Overall** |
|---|---|---|---|---|---|---|---|---|---|
| Proposed model | 96.92 | 93.38 | 87.86 | 82.72 | 84.46 | 86.51 | 87.38 | 90.82 | 87.84 |
|  | **0-2** | **4-6** | **8-13** | **15-20** | **25-32** | **38-43** | **48-53** | **60+** | **Overall** |
| *Agbo-Ajala & Viriri (2020)* Model | 94.59 | 82.35 | 85.43 | 76.34 | 87.16 | 67.86 | 62.69 | 91.09 | 81.58 |

the accuracy and reliability of facial expression-based age estimation is questionable as facial expressions can be influenced by various factors, including cultural background and personality. Relying on these estimates can result in false assumptions and unfair treatment of individuals. Therefore, it is crucial to consider the potential negative effects and ensure that all artificial intelligence systems developed to estimate social status from facial expression are developed with ethical principles such as transparency, accountability, and fairness in mind. It is also essential to ensure that such systems undergo rigorous testing and verification to minimize the risk of perpetuating societal biases and discrimination. Since this study only uses age-labeled datasets for security and system accessibility purposes, we believe that this study is outside of the ethical concerns mentioned above.

## CONCLUSIONS

People's aging processes are influenced by different factors such as lifestyle, exercise habits, genetics, environment, and weather conditions, which affect each individual differently. Therefore, age estimation from facial images is a difficult problem to solve. In the last two decades, artificial intelligence algorithms have been applied to solve this complex problem. Firstly, we presented the results of models from previously published studies and reviewed the performance values of the algorithms together with the number of parameters used. The most significant weakness in previous CNN architectures was the use of a large number of parameters, which increased the time and space complexity of the algorithms. For this reason, in this study, we presented a deep CNN model by trying to reduce the number of parameters and thus shorten the simulation time and lower the memory space used.

The dataset of the problem was obtained by combining the UTKFace dataset with the Facial-age dataset and then augmenting it. Reducing the number of parameters and increasing the dataset helped overcome the overfitting problem, reduced the running time of the algorithm, and minimized memory usage. The proposed CNN model has been compared with previous models and was shown to produce superior results in all respects. Although the number of parameters in the proposed deep CNN architecture is much less than that of existing architectures, it achieved an overall accuracy of 87.84%, outperforming previously published results. The experimental results also showed that the performance values specific to each age group are different from each other. This result is due to the fact that the distinguishing features in some age groups are similar to those in neighboring age groups. The results demonstrate that, in a given age group, the incorrectly predicted sample is placed in a group right next to it.

In this study, a series of experiments were conducted to observe the effect of resizing the original $200 \times 200$ pixel images to different resolutions ($100 \times 100$, $50 \times 50$, and $25 \times 25$) with appropriate layer numbers in the CNN model. Obviously, reducing the dimensions decreases the running time of the algorithm. However, performance values increased for $100 \times 100$ images and decreased for others. This can be explained by the fact that the images in general datasets deviate from their original size and decrease in visual clarity after being enlarged to $200 \times 200$ for consistency. A better result is obtained when the images are resized to $100 \times 100$, as it brings the images closer to their original forms. Further

reduction in size can be attributed to the loss of facial feature information, which degrades age estimation performance.

The performance of AI models is highly dependent on the structures of the models and the quantity and quality of datasets. Therefore, removing unimportant background elements from images can help improve the performance of age grouping algorithms. On the other hand, it would be valuable to prevent the problem of losing facial features while reducing the size of the images. In this way, both the performance values can be improved, and the simulation time will be shortened.

## ACKNOWLEDGEMENTS

This study is based on Ahmad Alsaleh's Master Thesis, Eskişehir Technical University.

### Funding

This work was supported by the Eskişehir Technical University Scientific Research Projects Commission (No. 21GAP084 and 22ADP144). The funders had no role in study design, data collection and analysis, decision to publish, or preparation of the manuscript.

### Grant Disclosures

The following grant information was disclosed by the authors:
Eskişehir Technical University Scientific Research Projects Commission: 21GAP084, 22ADP144.

### Competing Interests

The authors declare there are no competing interests.

### Author Contributions

- Ahmad Alsaleh conceived and designed the experiments, performed the experiments, analyzed the data, performed the computation work, prepared figures and/or tables, and approved the final draft.
- Cahit Perkgoz conceived and designed the experiments, analyzed the data, authored or reviewed drafts of the article, and approved the final draft.

### Data Availability

The code is available at GitHub and Zenodo: https://github.com/eaalsaleh/AE. 10.5281/zenodo.7792519.

eaalsaleh. (2023). eaalsaleh/AE: Initial release (v0.1.0). Zenodo. https://doi.org/10.5281/zenodo.7792519.

We used the following publicly available datasets:

- UTKFace dataset: https://susanqq.github.io/UTKFace/.
- Facial-age dataset: https://www.kaggle.com/datasets/frabbisw/facial-age.

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
