# Peer review of "A space and time efficient convolutional neural network for age group estimation from facial images"

_PeerJ Computer Science, doi:10.7717/peerj-cs.1395_

## Round 0.1 · original submission · Minor Revisions

This is an evolving field, and there are of course ethical considerations surrounding the prediction of social status from facial expression. It would benefit your article to add in some discussion on this issue.

Please address the concerns raised by the reviewers. Please include these changes in the manuscript and also attach a separate document explaining what changes were carried and where (page #, paragraph #), and that addresses the reviewer's suggestion/comment/concern. I will review them and if found satisfactory, I will accept the paper.

Thanks again for your interest in the journal.

Reviewer 1 ·

Basic reporting

- The paper is systematically organized into sections on Introduction, Materials & Methods, Results, Discussion and Conclusion which is useful for the readers. The quality of English is very good throughout.
- The authors have extensively cited previous work, which helps readers put this work into context with existing body of literature – especially in such a fertile field of research.
- The introduction quickly gets to the lacunae in the current body of research, which this manuscript intends to solve: accurate age estimation from facial images, with a reasonably small number of parameters.
- The authors have done a splendid job – they have made the manuscript very readable, with the crisp language, clear explanations, solid arguments, and interpretable simulation results. I congratulate them for this manuscript.

Experimental design

- In this manuscript, the authors use a deep CNN model with a reduced number of parameters, thereby reducing any potential overfitting.
- They utilize two publicly available datasets (plus augmentation), permitting the reader to check the performance of this model.
- They also observed the size reduction effect by resizing the original images to 1/4, 1/16 and 1/64 of their size and using appropriate layers.
- The authors also make comparison with existing Agbo-Ajala model, and demonstrate that the present method is superior to theirs, by a considerable (6%) margin.

Validity of the findings

- The algorithm appears really good to me, given their experimental results and comparison with state-of-the-art models. Although I do maintain some skepticism on how this model performs better by such a wide margin. Whether as an extension to this, or as a follow-up paper, it would be good to test against other datasets.

Additional comments

None

Reviewer 2 ·

Basic reporting

Review for paper – A space and time efficient convolutional neural network for age group estimation from facial images
Abstract – Abstract can be improved by adding additional details. What are the various factors that might impact facial age. AI technique is vague term instead expand on what AI techniques have been used to get most successful results. Now a days number of parameters are no more of issue because of big data and cheaper computational resources in existence by cloud services. Add more performance metrics in the abstract not limited to F-1 score.

Introduction – Introduction is neat and well structured. I would advise authors to add research related to recent years i.e., 2021/2022. Lot of references in this paper are taken from 2010-2015. Many advance strides of development happened in recent years and would like to see those in this paper.

Experimental design

Materials & Methods – Dataset is rightly chosen and formulas are correctly presented. No need to mention the early stages of CNN development that happened in 1968-1970. Focus more on recent development and novel methodologies that you proposed in this paper. Dataset is categorized into 8 parts why specifically into 8? what is the reasoning behind that ? The accuracy is mediocre not close to state of the art models accuracy and performance metrics. Research more on GANs and other deep learning techniques to improve the performance metrics and add a baseline model to compare your model results with that.

Validity of the findings

Answered above

---

## Round 0.2 · Minor Revisions

I reviewed the revisions you made and satisfied with the revision and response. I am glad to note the ethical issue discussion included in the paper.

The paper is technically ready to be published, but you need to do a thorough proofread first.

---

## Round 0.3 · accepted · Accept

Thanks for your interest in the journal.